# Resistance of Omicron subvariants BA.2.75.2, BA.4.6, and BQ.1.1 to neutralizing antibodies

Delphine Planas[1,2,16] ✉, Timothée Bruel [1,2], Isabelle Staropoli[1], Florence Guivel-Benhassine[1], Françoise Porrot [1], Piet Maes [3], Ludivine Grzelak[1], Matthieu Prot[4], Said Mougari[4], Cyril Planchais[5], Julien Puech[6], Madelina Saliba[6], Riwan Sahraoui[6], Florent Fémy[7], Nathalie Morel [8], Jérémy Dufloo [9], Rafael Sanjuán [9,10], Hugo Mouquet[5], Emmanuel André[11,12], Laurent Hocqueloux [13], Etienne Simon-Loriere [4], David Veyer[6,14], Thierry Prazuck[13,15], Hélène Péré[6,14,15] & Olivier Schwartz [1,2,16] ✉

Convergent evolution of SARS-CoV-2 Omicron BA.2, BA.4, and BA.5 lineages has led to the emergence of several new subvariants, including BA.2.75.2, BA.4.6. and BQ.1.1. The subvariant BQ.1.1 became predominant in many countries in December 2022. The subvariants carry an additional and often redundant set of mutations in the spike, likely responsible for increased transmissibility and immune evasion. Here, we established a viral amplification procedure to easily isolate Omicron strains. We examined their sensitivity to 6 therapeutic monoclonal antibodies (mAbs) and to 72 sera from Pfizer BNT162b2-vaccinated individuals, with or without BA.1/BA.2 or BA.5 breakthrough infection. Ronapreve (Casirivimab and Imdevimab) and Evusheld (Cilgavimab and Tixagevimab) lose antiviral efficacy against BA.2.75.2 and BQ.1.1, whereas Xevudy (Sotrovimab) remain weakly active. BQ.1.1 is also resistant to Bebtelovimab. Neutralizing titers in triply vaccinated individuals are low to undetectable against BQ.1.1 and BA.2.75.2, 4 months after boosting. A BA.1/BA.2 breakthrough infection increases these titers, which remains about 18-fold lower against BA.2.75.2 and BQ.1.1, than against BA.1. Reciprocally, a BA.5 breakthrough infection increases more efficiently neutralization against BA.5 and BQ.1.1 than against BA.2.75.2. Thus, the evolution trajectory of novel Omicron subvariants facilitates their spread in immunized populations and raises concerns about the efficacy of most available mAbs.

Successive sub-lineages of Omicron have spread worldwide since the identification of BA.1 in November 2021[1,2]. More than 80% of the population were infected by one or another Omicron subvariant in less than one year[3,4], without efficient protection against infection conferred by vaccination[5–7]. The incidence of breakthrough infections in vaccinated individuals has thus increased with Omicron[3,8]. All Omicron lineages exhibit considerable immune evasion properties. BA.1 and BA.2 contained about 32 changes in the spike protein, promoting immune escape and high transmissibility[9–11]. BA.5 was then predominant in many countries by mid-2022 and was responsible for a novel peak of contaminations[2,12]. BA.4 and BA.5 bear the same spike, with 4 additional modifications when compared to BA.2. The neutralizing activity of sera from COVID-19 vaccine recipients was further reduced against BA.4/BA.5 by about 3–5 fold compared to BA.1 and BA.2[12–15]. This reduced neutralization was associated with an abbreviated serum neutralization in triply

vaccinated individuals. the duration of neutralization was markedly shortened from 11.5 months with the ancestral D614G strain and 8 months with BA.1 to 5.5 months with BA.5[15]. Novel sub-variants with enhanced transmissibility rates, derived from either BA.2 or BA.4/BA.5, rapidly emerged and became prevalent in November-December 2022. Their geographical distribution is heterogeneous, but they carry an additional limited set of mutations in the spike. For instance, BA.2.75.2, derived from BA.2, was first noted in India and Singapore and comprises R346T, F486S, and D1199N substitutions[16–18]. BA.4.6 was detected in various countries, including USA and UK, and carries R346T and N658S mutations[19,20]. As of December 2022, BQ.1.1 became the main circulating lineage in many countries. It also carries the R346T mutation found in BA.2.75.2, along with K444T and N460K substitutions[21]. The R346T mutation has been associated with escape from monoclonal antibodies (mAbs) and from vaccine-induced antibodies[17,18,22]. This convergent evolution of the spike suggests that the different circulating SARS-CoV-2 sub-lineages faced a similar selective pressure, probably

exerted by preexisting or imprinted immunity[22,23]. With the large increase of breakthrough infections observed since Omicron emerged, "hybrid immunity" is also probably a main selective driver of SARS-CoV-2 immune evasion[24,25]. A characterization of these new viruses is needed to evaluate their potential impact.

A few recent articles and preprints reported an extensive escape of these Omicron subvariants to neutralization, studying sera from individuals who received three or four vaccine doses, including a bivalent booster[16,26–29]. Most of these studies were performed with lentiviral or VSV pseudotypes. Recombinant SARS-CoV-2 viruses carrying spikes from Omicron sublineages in an ancestral SARS-CoV-2 backbone were also generated[16], but they might behave somewhat differently than authentic isolates.

Here, we identify and use a highly permissive cell line to amplify BA.2.75.2, BA.4.6. and BQ.1.1 isolates. We analyze the sensitivity of these strains to approved mAbs, to sera from Pfizer BNT162b2 vaccine recipients, and to individuals with BA.1/BA.2 or BA.5 breakthrough infections.

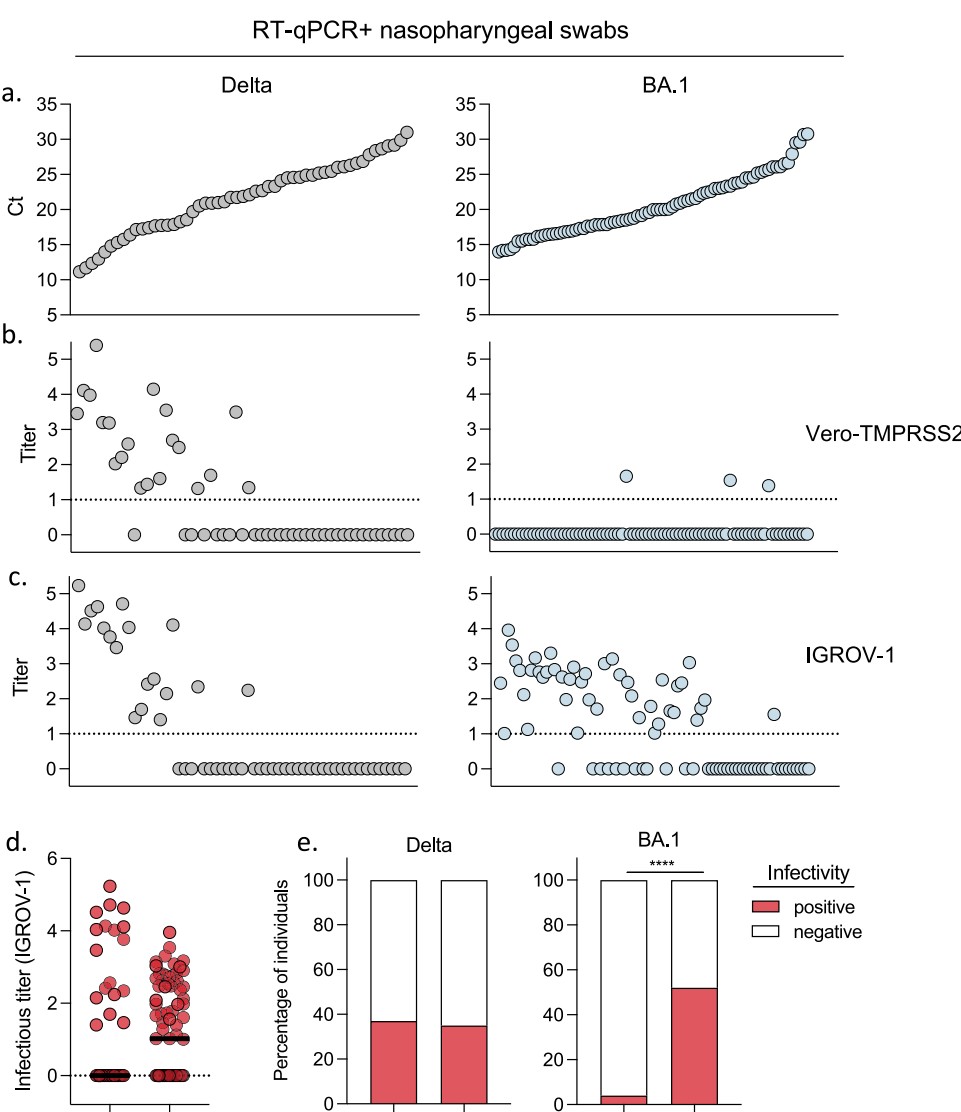

**Fig. 1 | Improved detection of infectious Omicron BA.1 in nasopharyngeal swabs using IGROV-1 cells.** A retrospective series of 135 RT+qPCR+ nasopharyngeal swabs from COVID-19 patients, harboring Delta (*n* = 53) or Omicron BA.1 (*n* = 82) variants was collected. **a** Viral RNA loads, measured by RT-qPCR. The samples were ranked from high to low viral RNA load (low to high Ct). Viral titers were measured in Vero-TMPRSS2 (**b**) and IGROV-1 cells (**c**). Delta and Omicron BA.1-positive samples are depicted in the left and right panels, respectively. **d** Comparison of infectious titers for Delta and BA.1 samples in IGROV-1 cells (left panel). **e** Percentage of samples harboring detectable infectious Delta (middle panel) or BA.1 virus (right panel) using Vero and IGROV-1 cells. A two-sided Chi-square test was performed ****$p$ < 0.0001. Source data are provided as a Source Data file.

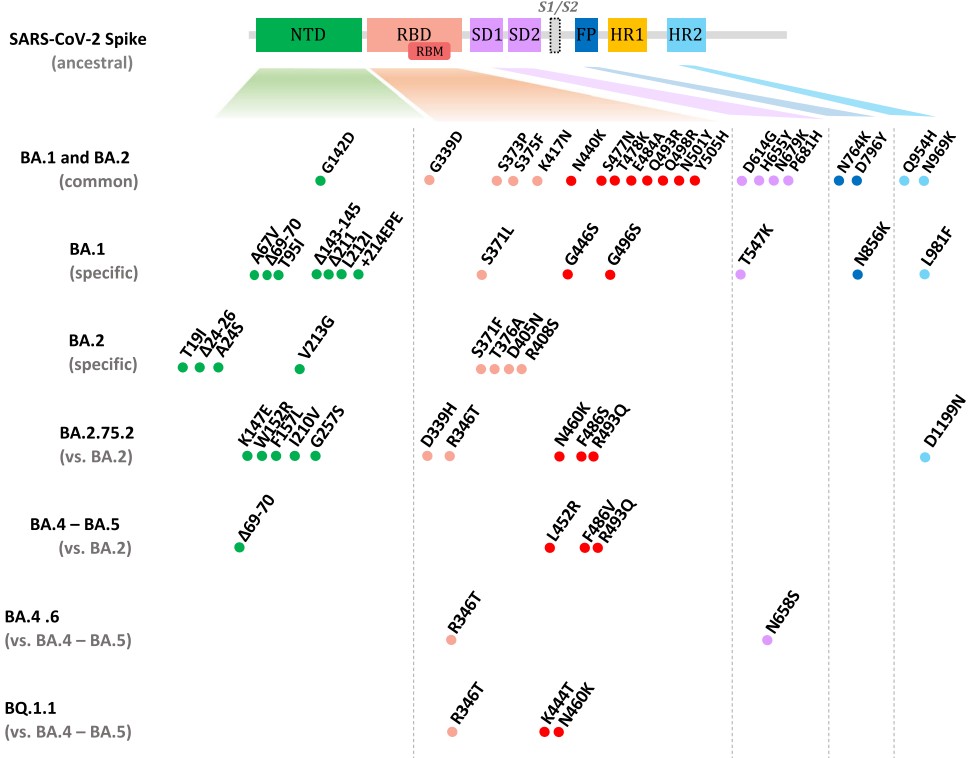

**Fig. 2 | Mutations present in the spike proteins of Omicron subvariants.** NTD, N-terminal domain; RBD, receptor binding domain; RBM, receptor binding motif; SD1, subdomain 1; SD2, subdomain 2; FP, fusion peptide; HR1, heptad repeat 1; HR2, heptad repeat 2. The BA.1 and BA.2 mutations are relative to the ancestral Wuhan sequence, the BA.2.75.2 mutations are relative to BA.2, the BA.4.6 and BQ.1.1 relative to BA.4/BA.5. Data are adapted from[21].

## Results

### Rapid isolation of Omicron subvariants with the IGROV-1 cell line

SARS-CoV-2 strains are classically isolated and amplified in Vero E6 or Vero-TMPRSS2+ cells. Vero cells are african green monkey kidney epithelial cells that were derived in the 1960s. They are defective in type-I interferon production and sensitive to many viral species[30]. However, upon serial passages in Vero E6 cells, SARS-CoV-2 may acquire adaptive spike mutations, with modification or deletion of the furin-like cleavage site, resulting in phenotypic changes in plaque assays[31]. Omicron isolates are growing less efficiently in Vero E6 and Vero-TMPRSS2+ cells than previous SARS-CoV-2 variants, probably because Omicron relies more on endocytic proteases and less on TMPRSS2 than other variants[32,33]. This may explain why infectious viral loads measured in nasopharyngeal swabs from Omicron-infected individuals appeared lower than those infected with Delta, despite an enhanced transmissibility of Omicron[34]. We thus sought another cell line that may be more adapted to isolation and replication of Omicron subvariants than Vero cells. To this aim, we screened a panel of cells and observed that IGROV-1 cells were highly permissive to Omicron. IGROV-1 cells originated from an ovarian carcinoma and were established in 1985[35]. IGROV-1 cells naturally express low levels of ACE2 and TMPRSS2, as assessed by flow cytometry (Extended data Fig. 1).

We compared the permissibility of Vero-TMPRSS2+ and IGROV-1 cells to Omicron and Delta. We titrated infectious viral loads in nasopharyngeal swabs from 53 Delta and 81 Omicron (BA.1) infected individuals collected at the Hôpital Européen Georges Pompidou (HEGP) in Paris. The characteristics of the patients (age, sex, days post onset of symptoms, vaccination status) appear in Supplementary Table 1. Nasopharyngeal swabs were serially diluted and incubated with either Vero-TMPRSS2+ or IGROV-1 cells. After 48 h, cells were stained with an anti-SARS-CoV-2 N monoclonal antibody.

Foci of infected cells were scored with an automated confocal microscope. A representative experiment with Delta and Omicron positive samples demonstrated a high sensitivity of IGROV-1 to Omicron (extended data Fig. 2).

The 134 samples were ranked according to their viral RNA levels measured by RT-qPCR, from low to high Ct (Fig. 1a). With Delta-positive samples, there was no major difference in infectious viral titers calculated with Vero-TMPRSS2+ or IGROV-1 cells which inversely correlated with Ct (Fig. 1b, c). We did not detect infectious virus in samples with Ct > 27. About 35% of Delta positive samples carried infectious virus (Fig. 1d, e). The situation was different with Omicron BA.1 positive samples. We did not detect Omicron-infected Vero-TMPRSS2+ cells, even in samples with low Ct, at this early time-point (48 h). In contrast, 52% of the samples from Omicron-infected individuals were positive when titrated on IGROV-1 cells (Fig. 1c–e), confirming that these cells are particularly sensitive to Omicron BA.1.

We next isolated BA.4.6 and BQ.1.1 variants from nasopharyngeal swabs collected at HEGP using IGROV-1 cells. As with BA.1, numerous foci of infected cells were detected at 2 days post-infection (p.i.) and supernatants were harvested at days 2 or 3 p.i., yelding high titers with the S-Fuse reporter cells. S-Fuse cells form syncytia and become GFP + upon infection, allowing overnight measurement of viral infectivity and neutralizing antibody activity[36,37]. Sequences of the variants after one passage on IGROV-1 cells identified BA.4.6 and BQ.1.1 (Pango lineage B.1.1.529.4.6 and B.1.1.529.5.3.1.1.1.1.1.1, respectively according to Nextstrain, GISAID accession ID: BA.4.6: EPI_ISL_15729633 and BQ.1.1: EPI_ISL_15731523), indicating that no adaptative mutations were generated during this short culture period. As expected, BA.4.6 included R346T and N658S mutations[19,20] and BQ.1.1 carried R346T, K444T, and N460K substitutions[21]. The spike mutations in the main Omicron subvariants are depicted Fig. 2.

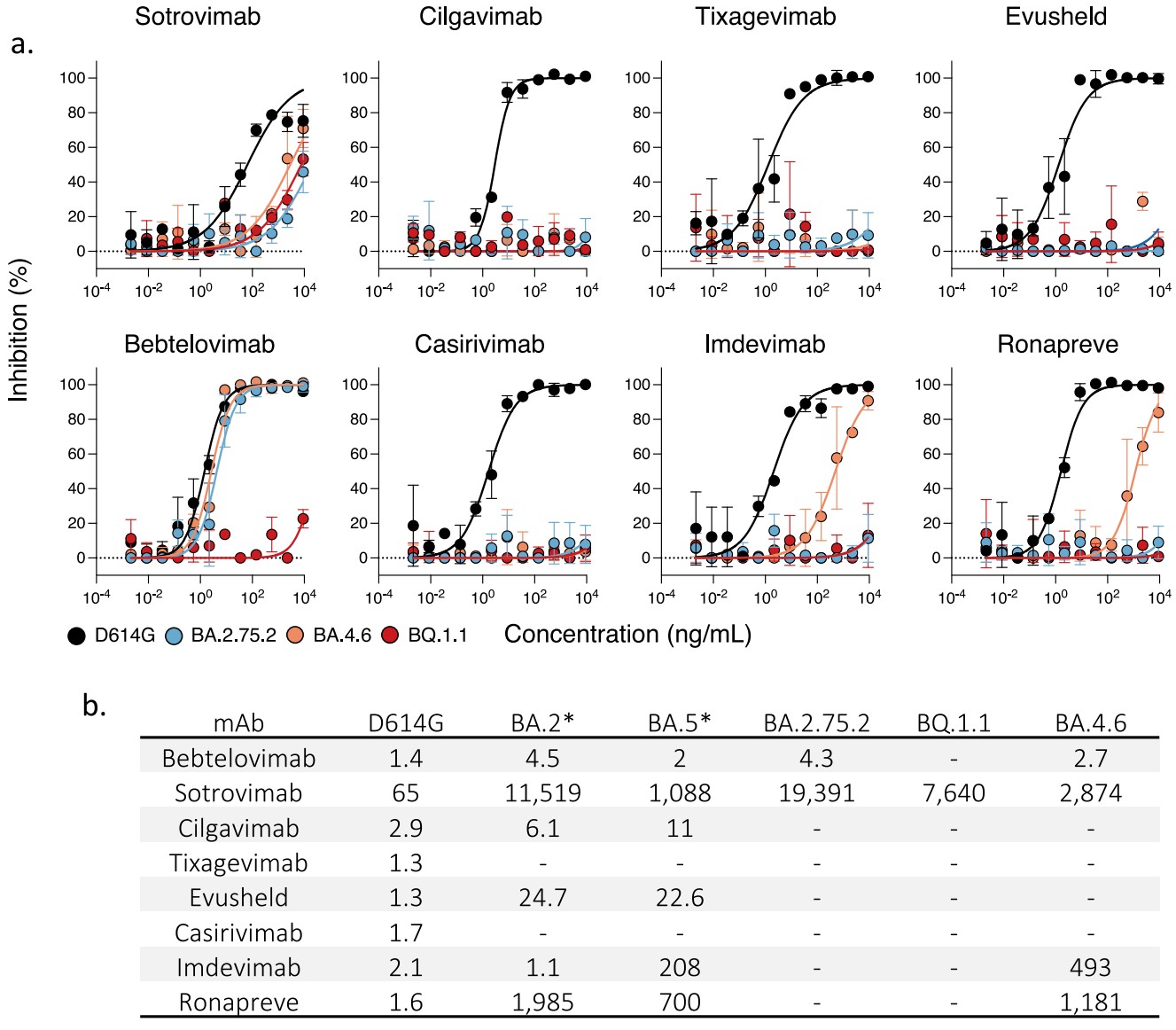

**Fig. 3 | Neutralization activity of therapeutic monoclonal antibodies against BQ.1.1, BA.2.75.2, and BA.4.6. a.** Neutralization curves of monoclonal antibodies. Dose–response analysis of the neutralization by the indicated antibodies or their clinical combinations. Evusheld: Cilgavimab and Tixagevimab. Ronapreve: Casirivimab and Imdevimab. Data are mean ± s.d. of $n = 2$ independent experiments. **b** IC50 values in ng/ mL for each antibody against the indicated viral strains. *ED50 against BA.2 and BA.5 are from[49]. Source data are provided as a Source Data file.

We also isolated a BA.2.75.2 variant from a nasopharyngeal swab from the National Reference Center of UZ/KU Leuven (Belgium). The virus was initially amplified by two passages on Vero E6 cells, but the resulting viral titers were low. We thus performed one supplementary passage on IGROV-1 cells, which significantly increased the titers to $4 \times 10^5$ pfu/ml in 48 h. Sequencing of the virus confirmed the presence of BA.2.75.2 (Pango lineage B.1.1.529.2.75.2, according to Nextstrain, GISAID accession ID: E EPI_ISL_15731524). When compared to BA.2.75, the BA.2.75.2 spike protein contained 3 additional mutations, R346T and F486S in the RBD, and D1199N in the HR2 (Heptad Repeat 2) region, located in the S2 domain and involved in fusion (Fig. 2).

Syncytia were observed in BA.2.75.2, BA.4.6., and BQ.1.1-infected S-Fuse cells (Extended Data Fig. 3). The three variants generated syncytia of similar size, that were smaller than those formed by the ancestral D614G strain (Extended Data Fig. 3). It will be worth further

examining whether other Omicron subvariants may display different fusogenic potential in different cell types.

Altogether, these results show that IGROV-1 cells are highly sensitive to Omicron. They allow a rapid titration of infectivity present in nasopharyngeal swabs from infected individuals, as well as a one-passage amplification of Omicron subvariants. Future work will help determining the underlying cellular mechanisms and whether entry or other steps of the viral cycle are facilitated in IGROV-1 cells.

**Neutralization of BA.2.75.2, BA.4.6. and BQ.1.1 by approved monoclonal antibodies**

Several anti-spike monoclonal antibodies (mAbs) are used as pre-exposure prophylaxis (PrEP) or post-exposure therapy in individuals at risk for severe disease[38]. These mAbs belong to the four main classes of anti-RBD antibodies which are defined by their binding site[38,39].

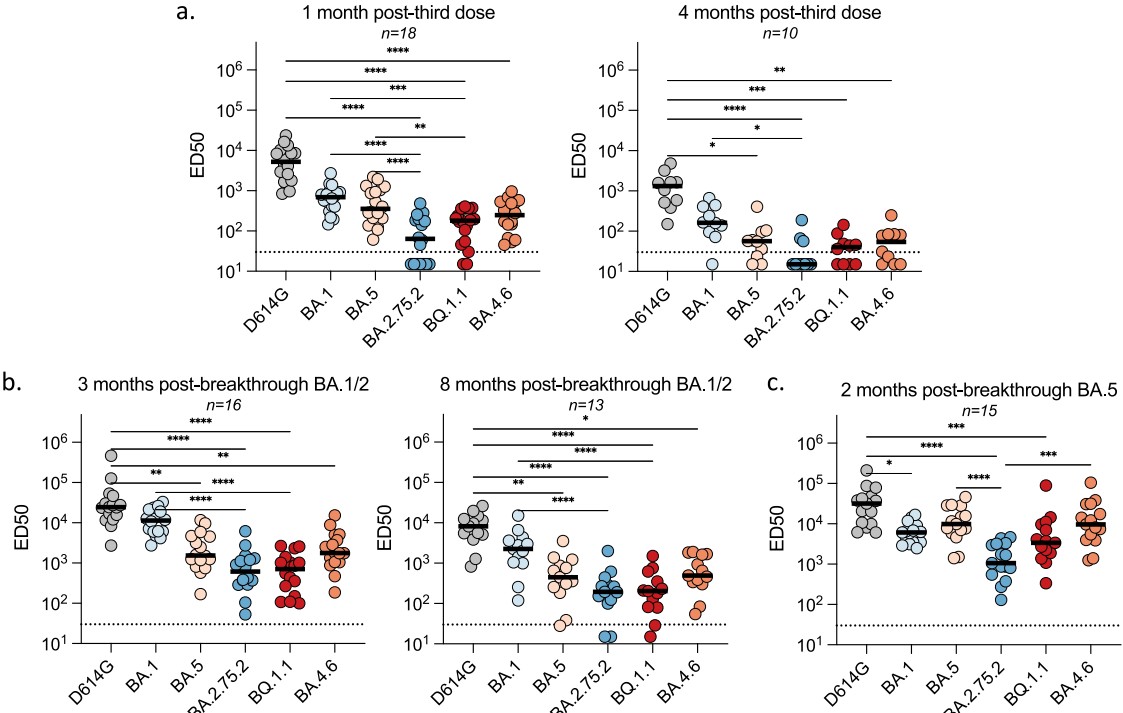

**Fig. 4 | Sensitivity of SARS-CoV-2 D614G and Omicron subvariants to sera from vaccinated, or infected-then-vaccinated individuals.** Neutralization titers of the sera against the indicated viral variants are expressed as ED50. **a.** Neutralizing activity of sera from individuals vaccinated with 3 doses of Pfizer vaccine. Sera were sampled at 1 month (left panel; $n = 18$) and 4 months (right panel; $n = 10$) after the third dose. **b** Neutralizing activity of sera from Pfizer-vaccinated recipients after BA.1/BA.2 breakthrough infection. Sera were sampled about 3 months (left panel; $n = 16$) and 8 months (right panel; $n = 13$) after the breakthrough. **c** Neutralizing activity of sera from Pfizer-vaccinated recipients after BA.5 breakthrough infection. Sera were sampled about 2 months after the breakthrough ($n = 15$). The dotted line indicates the limit of detection (ED50 = 30). Black lines represent the median values. Two-sided Friedman test with Dunn's test for multiple comparisons was performed between each viral strain at the different time points; *$p < 0.05$; **$p < 0.001$; ***$p < 0.0001$; ****$p < 0.0001$. 1 month post-third dose: D614G versus BA.2.75.2, $P < 0.0001$; D614G versus BQ.1.1, $P < 0.0001$; D614G versus BA.4.6,

$P < 0.0001$; BA.1 versus BA.2.75.2, $P < 0.0001$; BA.1 versus BQ.1.1, $P = 0.0005$; BA.5 versus BA.2.75.2, $P < 0.0001$; BA.5 versus BQ.1.1, $P = 0.0033$. 4 months post-third dose: D614G versus BA.5, $P = 0.0152$; D614G versus BA.2.75.2, $P < 0.0001$; D614G versus BQ.1.1, $P = 0.0003$; D614G versus BA.4.6, $P = 0.0025$; BA.1 versus BA.2.75.2, $P = 0.0123$. 3 months post-breakthrough BA.1/2: D614G versus BA.5, $P = 0.0034$; D614G versus BA.2.75.2, $P < 0.0001$; D614G versus BQ.1.1, $P < 0.0001$; D614G versus BA.4.6, $P = 0.0071$; BA.1 versus BA.2.75.2, $P < 0.0001$; BA.1 versus BQ.1.1, $P < 0.0001$. 8 months post-breakthrough BA.1/2: D614G versus BA.5, $P = 0.0024$; D614G versus BA.2.75.2, $P < 0.0001$; D614G versus BQ.1.1, $P < 0.0001$; D614G versus BA.4.6, $P = 0.0173$; BA.1 versus BA.2.75.2, $P < 0.0001$; BA.1 versus BQ.1.1, $P < 0.0001$. 2 months post-breakthrough BA.5: D614G versus BA.1, $P = 0.0192$; D614G versus BA.2.75.2, $P < 0.0001$; D614G versus BQ.1.1, $P = 0.0009$; BA.5 versus BA.2.75.2, $P < 0.0001$; BA.2.75.2 versus B.4.6, $P = 0.0002$. Source data are provided as a Source Data file.

Prophylaxis based on Ronapreve (Imdevimab + Casirivimab) or Evusheld (Cilgavimab + Tixagevimab) cocktails provided about 80% protection against symptomatic infection[40,41]. Post-infection treatment with Xevudy (Sotrovimab) reached 85% efficacy in preventing COVID-19-related hospitalization or death[38,42]. However, Omicron BA.1, BA.2, and BA.5 escaped neutralization from a large part of these mAbs, leading to changes in treatment guidelines[43,44]. As of mid-2022, Ronapreve and Sotrovimab were no longer approved and a double dose of Evusheld was recommended. Bebtelovimab is another potent mAb, similarly effective against ancestral strains and BA.1 and BA.2[45], currently only available in United States[46].

We thus assessed with the S-Fuse assay the sensitivity of BA.2.75.2, BA.4.6. and BQ.1.1 to mAbs that are currently authorized (Cilgavimab, Tixagevimab and Bebtelovimab) or were withdrew because of Omicron escape (Sotrovimab, Casirivimab, and Imdevimab). As controls, we included the ancestral D614G strain (Fig. 3a, b). Cilgavimab and Tixagevimab, alone or in combination, as well as Casirivimab, lost any neutralization activity against the three Omicron variants. Imdevinab inhibited BA.4.6 (IC50 493 ng/ml) but was inactive against BA.2.75.2 and BQ.1.1. Bebtelovimab was efficient against BA.4.6 and BA.2.75.2 (IC50 2.7 and 4.3 ng/ml, respectively) but did not neutralize BQ.1.1. Sotrovimab was the only mAb active, albeit weakly, against BA.2.75.2, BA.4.6. and BQ.1.1. With Sotrovimab, the IC50s ranged from 2,874 to

19,391 ng/ml, which represents a 45-to-300-fold increase compared to D641G.

These results demonstrate that the prevalent BA.2.75.2 and BQ.1.1 strains are resistant or weakly sensitive to currently approved mAbs.

### Cohort design

We collected 72 sera from a cohort of 35 health-care workers, in Orleans, France. We previously studied the ability of some of these sera to neutralize Alpha, Beta, Delta, Omicron BA.1 and BA.5 variants[11,15]. The characteristics of the participants are indicated in Supplemental Table 2. The participants, that were not previously infected at the time of inclusion, received two doses of Pfizer BNT162b2 vaccine within an interval of 21–28 days and a booster dose 164 to 314 days later. 31 out of 35 individuals experienced a pauci-symptomatic breakthrough Omicron infection 60 to 359 days after the third injection. Screening by PCR or whole viral genome sequencing identified the Omicron sub-variant responsible for the breakthrough infection. A first group of 16 individuals was infected between December 2021 and mid-June 2022, a period when BA.1 and BA.2 were successively dominant in France[47]. A second group of 15 individuals was infected between July and October 2022 and was positive for BA.5. We did not have access to their samples prior to their BA.5 breakthrough infection. The days of vaccination,

breakthrough infection and sampling are displayed in Supplemental Table 2.

### Sensitivity of BA.2.75.2, BA.4.6. and BQ.1.1 to sera from vaccinees

We asked whether vaccine-elicited antibodies neutralized the novel Omicron subvariants. Eighteen individuals were analyzed early (1 month post third dose) and among them, ten individuals that did not experience a breakthrough infection were analyzed at a later time-point (4 months post third dose). We measured the potency of their sera against BA.2.75.2, BA.4.6., and BQ.1.1. We used as controls the D614G ancestral strain (belonging to the basal B.1 lineage), as well as BA.1 and BA.5 (Fig. 4a). We calculated the ED50 (Effective Dose 50%) for each combination of serum and virus. One month after the booster dose, ED50 were high for D614G (ED50 of $5 \times 10^3$) and were decreased by 8- and 15-fold for BA.1 (ED50 of $7 \times 10^2$) and BA.5 (ED50 of $3 \times 10^2$) respectively, confirming the antibody escape properties of these previous sublineages. With BA.4.6. and BQ.1.1, the ED50 were low and within the range of those observed with the parental BA.5 strain. BA.2.75.2 neutralization titers were even lower (11-fold lower than BA.1). A similar trend was observed at a later time-point. Neutralization was reduced against all strains, highlighting the declining humoral response[11,15]. The neutralizing activity was either undetectable or barely detectable against BA.2.75.2, BA.4.6., and BQ.1.1 (Fig. 3a).

Altogether, these results indicate that the prevalent Omicron subvariants are poorly or not neutralized by vaccinees' sera sampled 4 months after a third vaccine dose.

### Impact of BA.1/BA.2 breakthrough infections on neutralization of Omicron subvariants

We then examined the impact of BA.1/BA.2 breakthrough infections on the cross-neutralizing activity of serum antibodies. Eighteen individuals were analyzed about 3 months post-infection (median 84 days; range 44–109 days). Among them, 13 individuals were resampled about 8 months (median 234 days; range 142–289 days) after infection to evaluate the evolution of the humoral response. After 3 months, a strong augmentation of neutralization against D614G and BA.1 was observed, with ED50 above $10^4$ (Fig. 4b). Compared to BA.1, the Nab titers were reduced by about 7-fold against BA.5 and BA.4.6 (ED50 of $1.5 \times 10^3$ and $1.8 \times 10^3$, respectively) and reduced by 18-fold against BA.2.75.2 and BQ.1.1 (ED50 of $6 \times 10^2$ and $7 \times 10^2$, respectively). Neutralizing titers differently declined depending on the viral isolate. Eight months after infection, titers remained high against D614G and BA.1 (ED50 of $8 \times 10^3$ and $3 \times 10^3$, respectively). The decline was stronger against BA.5 and BA.4.6 (ED50 of $4 \times 10^2$) and even more marked against BA.2.75.2 and BQ.1.1 (ED50 of $2 \times 10^2$). Therefore, post-vaccination infection by BA.1/BA.2 led to an increase in Omicron-specific neutralizing antibody titers, with disparities between variants. The anti-BA.1 response was higher than against BA.5 and BA.4.6, whereas BA.2.75.2 and BQ.1.1 were less sensitive to neutralization.

### Impact of BA.5 breakthrough infections on neutralization of Omicron subvariants

The distinct neutralization profile of BA.2.75.2, BA.4.6. and BQ.1.1 after BA.1/BA.2 infection led us to examine the consequences of a BA.5 breakthrough infection on neutralization. We assessed the sera of fifteen individuals, about two months after BA.5 infection (median 50 days; range 12–127 days). As for BA.1/BA.2 breakthrough infection, we observed a strong augmentation of neutralization against D614G, with ED50 reaching $3 \times 10^4$ (Fig. 4c). The neutralization of BA.5 variants (BA.5 and BQ.1.1) was high (ED50 of $10^4$) and 10-fold lower for the BA.2-derived BA.2.75.2 strain (ED50 of $1 \times 10^3$). The neutralization activity against BA.1 was less potent after a BA.5 infection than after a BA.1/BA.2 infection (ED50 of $6 \times 10^3$ and $1 \times 10^4$, respectively) (Fig. 4b, c).

Altogether, these results indicate that a BA.5 breakthrough infection triggers a better neutralization of viral isolates of the BA.5 lineage than BA.1/BA.2-derived strains. Conversely, A BA.1/BA.2 breakthrough infection favors neutralization of BA.1 and BA.2 derived strains, relative to the BA.5 lineage.

## Discussion

We report here a simple method to isolate and grow Omicron strains. We identify IGROV-1 cells as being highly permissive to Omicron, through reasons that remain to be determined. Omicron is less replicative and fusogenic than Delta in various human cell lines including Vero, Calu-3, A549-ACE2, HeLa-ACE2/TMPRSS2 and U2OS-derived S-Fuse cells[11,15,48,49]. Omicron strains inefficiently use TMPRSS2, which promotes viral entry through plasma membrane fusion, with greater dependency on endocytic entry[32,33]. Several lines of evidence indicate that the evolution of Omicron sublineages towards increased transmissibility is associated with greater fitness in human primary cells. BA.1 potently replicates in nasal epithelial cultures[32]. BA.4 and BA.5 replicate more efficiently than BA.2 in alveolar epithelial cells and are more fusogenic[50]. BA.2.75.2 growth efficiency in alveolar epithelial cells and spike-mediated fusion in Calu-3 cells are also higher than those of BA.2[51,52]. We did not observe an enhanced cell-cell fusogenicity of BA.2.75.2, BA.4.6. and BQ.1.1 compared to BA.5, at least in S-Fuse cells, but it will be worth further examining viral fitness and fusion of these strains in IGROV-1 or primary cells.

Our results show that IGROV-1 cells recapitulate the permissibility of primary human nasal or alveolar cells to Omicron strains. Future work will help understanding viral entry pathways and replication in IGROV-1 cells. Whatever the underlying mechanisms, these cells proved useful to amplify BA.2.75.2, BA.4.6., and BQ.1.1 in a single passage, avoiding or minimizing the risk of selection of culture adaptative mutations[31]. IGROV-1 cells are also sensitive to previous SARS-CoV-2 variants. Combining viral isolation in IGROV-1 cells with the S-Fuse neutralization assay provides a rapid procedure to evaluate the properties of novel and forthcoming SARS-CoV-2 variants of concern.

We demonstrate that the currently approved or recently withdrawn therapeutic mAbs lost most of their neutralization potential against these Omicron subvariants. Evusheld no longer neutralized BA.2.75.2, BA.4.6. and BQ.1.1. Bebtelovimab was active against BA.2.75.2 and BA.4.6. but not against BQ.1.1. This fits with the observation that the K444 residue, mutated in BQ.1.1 (K444T) but not in BA.2.75.2 and BA.4.6, is important for Bebtelovimab activity[45]. Ronapreve was active against BA.4.6 but not against the prevalent BA.2.75.2 and BQ.1.1 isolates. Sotrovimab retained a relatively low neutralization activity against all strains, with IC50 ranging from 3 to more than 9 μg/ml. Sotrovimab also displays non-neutralizing antiviral activities, including ADCC[53,54]. Sotrovimab remains clinically active against BA.2[55]. It will be of interest determining whether Sotrovimab could maintain some activity in vivo against these novel Omicron subvariants, despite reduced neutralization. This will help addressing the debate on the need to reassess WHO's therapeutics and COVID-19 living guideline on mAbs[56]. Overall, our results are in line with a recent preprint using lentiviral pseudotypes[22] and raise important concerns regarding the prophylactic and therapeutic administration of currently approved mAbs. Novel mAbs, with broad cross-neutralizing activities and inhibiting most of Omicron sublineages have been identified[22,57] and are warranted to extend the arsenal of mAb-based treatments.

We report that sera from individuals who had received three doses of COVID-19 Pfizer BNT162b2 vaccine displayed reduced neutralization activity against the Omicron subvariants. One month after a first booster, ED50 displayed a 10- to 80-fold decrease compared to the ancestral D614G strain. At 4 months post vaccination, neutralization was undetectable for BA.2.75.2 and slightly above background for BA.4.6. and BQ.1.1. These

results suggest an abbreviated efficacy of Pfizer BNT162b2 vaccine against the three variants, extending our previous results with BA.1 and BA.5[15]. The advantage of administrating monovalent or bivalent boosters is under scrutiny[26,28,58,59]. Preliminary preprints using lentiviral pseudotypes indicated that BA.5, BA.4.6, or BA.2.75 titers were comparable after monovalent or BA.5 bivalent boosters[26,28]. In contrast, when using tests based on recombinant SARS-CoV-2 infectious virus carrying spikes from different Omicron sublineages, it was observed that bivalent mRNA booster may broaden humoral immunity[27]. These discrepancies may be due to differences in experimental systems, the delay between booster administration and blood sampling, and/or variation in immune imprinting across cohorts. Future work with authentic field isolates and well-characterized sera, combined with real-world vaccine efficacy data[60], will help characterizing the interest of bivalent vaccines against Omicron subvariants.

We observed a dichotomy of the neutralizing response after BA.1/BA.2 or BA.5 breakthrough infection in vaccinated individuals. In both cases, the Nabs were particularly high against D614G, highlighting the role of immune imprinting in anamnestic responses[57,61]. Although increased neutralization occurs against the breakthrough strain, the vast bulk of the response likely corresponds to boosting of rarer cross-reactive antibodies related to imprinting with ancestral strain vaccines or infections[61]. However, after BA.1/BA.2 infection, sera also potently neutralized BA.1 but there was a 6 to 18-fold reduction in efficacy against BA.2.75.2, BA.4.6. and BQ.1.1. Conversely, after BA.5 breakthrough infection, titers were higher against BA.5-derived variants than against BA.1 or BA.2.75.2. It has been reported that vaccinated individuals infected during the first Omicron wave showed enhanced immunity against earlier variants but reduced nAb potency and T cell responses against Omicron[62]. This was not exactly the case in our study, suggesting that in addition to imprinted memory, other mechanisms such as the generation of responses targeting novel antigens could be operative. Here, we confirm that hybrid immunity, generated in vaccinated individuals after a breakthrough infection, leads to higher antibody titers regardless of the viral variants. Future investigation of the B cell repertoire in individuals with and without breakthrough infections will help deciphering the drivers of immune evasion in current or future variants.

Besides the RBD, antibodies targeting other regions of the spike, such as the NTD or S2 region may also broaden the humoral response[63,64]. The interval between prior SARS-CoV-2 infection and booster vaccination impacts magnitude and quality of antibody and B cell responses[65]. This raises important questions regarding the frequency of booster doses, particularly in the presence of Omicron variants with greater immune evasion properties.

There are several limitations to our study, notably the limited number of individuals analyzed. However, the differences between strains and categories of individuals were sufficiently marked to reach statistical significance. We did not consider the effect of innate and cellular immunity on BA.2.75.2, BA.4.6., and BQ.1.1 strains. We focused our work on Pfizer vaccine recipients and did not assess the neutralization conferred by a fourth dose. We did not characterize other Omicron subvariants, such as XBB, a recombinant virus between two Omicron strains (BJ.1 and BM.1.1). We analyzed the impact of breakthrough infections up to 8 months after BA.1/BA.2 infection and only 2 months after BA.5 infection. Future studies will help evaluating long-term immune responses to Omicron subvariants after infection and/or vaccination.

In summary, we show here that the few convergent mutations present in the spike of BA.2 or BA.5 subvariants led to resistance to most of available therapeutic mAbs and strongly impaired the efficacy of vaccine-elicited antibodies. Breakthrough infections in triply vaccinated individuals stimulate cross-neutralizing responses with distinct efficacy depending on the variant responsible for the infection. The evolution trajectory of the novel Omicron subvariants likely reflects their continuous circulation in immunized populations.

## Methods

Our research fulfills all relevant ethical requirements. The ABCOVID study was approved by the Ile-de-France IV ethical committee. The study with nasopharyngeal swabs from infected individuals was carried out in accordance with the Declaration of Helsinki and was evaluated by the ethics committee "Comité d'éthique de la recherche AP-HP Center" affiliated to the AP-HP. An informed consent was obtained from all participants

No statistical methods were used to predetermine sample size and the experiments were not randomized. The investigators were not blinded. Sex or gender analysis was not performed due to the limited number of participants.

### Cohorts

**Serum from vaccinated and BA.1/2 and BA.5 breakthrough infected individuals (Orléans cohort).** A prospective, monocentric, longitudinal, interventional cohort clinical study (ABCOVID) is conducted since 27 August 2020 with the objective to study the kinetics of COVID-19 antibodies in patients with confirmed SARS-CoV-2 infection (NCT04750720). A sub-study aimed to describe the kinetic of neutralizing antibodies after vaccination. The cohort was previously described[11,15]. These publications[11,15] and the present results are the primary outcomes of this clinical study. Anti-N antibodies were measured at the time of enrollment to exclude individuals infected before vaccination. Fifteen individuals were enrolled in November 2022 after BA.5 breakthrough infection, without known history of previous infection. This study was approved by the Ile-de-France IV ethical committee. At enrollment, written informed consent was collected and participants completed a questionnaire covering sociodemographic characteristics. Virological findings (SARS-CoV-2 RT−qPCR results, date of positive test, screening, or sequences results) and data related to SARS-CoV-2 vaccination (brand product, date of first, second, third, and fourth vaccination) were also collected.

**Nasopharyngeal swabs from infected individuals (Hôpital Européen Georges Pompidou).** 134 nasopharyngeal swabs collected for standard care between December 2, 2021 and January 5, 2022 were retrospectively analyzed to investigate Delta and Omicron BA.1 replication. This study was carried out in accordance with the Declaration of Helsinki with no sampling addition to usual procedures and was evaluated by the ethics committee "Comité d'éthique de la recherche AP-HP Center" affiliated to the AP-HP (Assistance publique des Hopitaux de Paris; IRB registration # 00011928). An informed consent was obtained from all participants. Swab specimens were collected for standard diagnostic following medical prescriptions in HEGP and stored at −80 °C prior to infectivity measurements and viral isolations.

### Virus strains

The reference D614G strain (hCoV-19/France/GE1973/2020) was supplied by the National Reference Center for Respiratory Viruses hosted by Institut Pasteur and headed by S. van der Werf. This strain was obtained through the European Virus Archive goes Global (Evag) platform, a project that has received funding from the European Union's Horizon 2020 research and innovation program under grant agreement no 653316. The BA.2.75.2 strain was isolated and sequenced by the NRC UZ/KU Leuven (Leuven, Belgium). BQ.1.1 and BA.4.6 were isolated from a nasopharyngeal swab of individuals attending the emergency room of Hôpital Européen Georges Pompidou (HEGP; Assistance Publique, Hôpitaux de Paris). The swabs were sequenced by the laboratory of Virology of HEGP. All patients

provided informed consent for the use of the biological materials. The variant strains were isolated from nasopharyngeal swabs using Vero E6 or IGROV-1 cells. Viral strains were amplified by one or two passages on Vero cells. Only one passage was necessary for the amplification on IGROV-1 cells. Supernatants were harvested 2 or 3 days after viral exposure. Titration of viral stocks was performed on Vero E6 cells, with a limiting dilution technique enabling the calculation of the median tissue culture infectious dose or on S-Fuse cells. Viral supernatants were sequenced directly from the nasopharyngeal swabs, and after their isolation and amplification on Vero or IGROV-1 cells. For sequencing, we used an untargeted metagenomic sequencing approach with ribosomal RNA (rRNA) depletion. Briefly, RNA was extracted with the QIAamp Viral RNA extraction kit (Qiagen), with the poly-A RNA carrier provided. Prior to library construction, carrier RNA and host rRNA were depleted using oligo (dT) and custom probes respectively. The RNA resulting from selective depletion was used for random-primed cDNA synthesis using the SuperScript IV RT (Invitrogen). Second-strand cDNA was generated using Escherichia coli DNA ligase, RNAse H and DNA polymerase (New England Biolabs) and purified using Agencourt AMPure XP beads (Beckman Coulter). Libraries were then prepared using the Nextera XT kit and sequenced on an Illumina NextSeq500 platform (2 × 75 cycles). Reads were assembled using megahit v1.2.9. The sequences were deposited on GISAID (D614G: EPI_ISL_414631; BA.1 ID: EPI_ISL_6794907; BA.5 ID: EPI_ISL_13660702; BA.2.75.2 ID: EPI_ISL_15731524; BQ.1.1 ID: EPI_ISL_15731523; BA.4.6 ID: EPI_ISL_15729633)[36,66].

### Cell lines
IGROV-1 cells were from the NCI-60 cell line panel and have been authenticated[67]. Vero E6 and Vero-TMPRSS2 were described previously[36,66]. 293T (CRL-3216) and U2OS (Cat# HTB-96) cells were obtained from ATCC. S-Fuse cells have been described previously[36,37].

### S-Fuse neutralization assay
U2OS-ACE2 GFP1–10 or GFP 11 cells, also termed S-Fuse cells, become GFP+ when they are productively infected by SARS-CoV-2[36,37]. Cells tested negative for mycoplasma. Cells were mixed (ratio 1:1) and plated at $8 \times 10^3$ per well in a µClear 96-well plate (Greiner Bio-One). The indicated SARS-CoV-2 strains were incubated with serially diluted monoclonal antibodies or sera for 15 min at room temperature and added to S-Fuse cells. Sera were heat-inactivated for 30 min at 56 °C before use. 18 h later, cells were fixed with 2% PFA (Electron microscopy cat# 15714-S), washed and stained with Hoechst (dilution of 1:1000, Invitrogen, Invitrogen cat# H3570). Images were acquired using an Opera Phenix high-content confocal microscope (PerkinElmer). The GFP area and the number of nuclei were quantified using the Harmony software (PerkinElmer). The percentage of neutralization was calculated using the number of syncytia as value with the following formula:   $100 \times (1 - (\text{value with serum} - \text{value in 'non-infected'})/(\text{value in 'no serum'} - \text{value in 'non-infected'}))$. Neutralizing activity of each serum was expressed as the half maximal effective dilution (ED50). ED50 values (in ng/ml for monoclonal antibodies and in dilution values −i.e titers−for sera) were calculated with a reconstructed curve using the percentage of neutralization at each concentration. Of note, we previously reported a correlation between neutralization titers obtained with the S-Fuse reporter assay and a pseudovirus neutralization assay[68].

### Nasopharyngeal swabs infectivity
Vero and IGROV-1 cells were plated at 30,000 cells per well in a mClear 96-well plate (Greiner Bio-One). The nasopharyngeal swabs were added to the Vero or IGROV-1 cells at serial dilutions from 1:10 to 1:31 250 as described previously[66]. 48 h later, cells were fixed with 2% PFA (Electron microscopy cat# 15714-S). Cells were washed and

stained intracellularly in 0.05% saponin with the anti-SARS-CoV-2 nucleoprotein (N) antibody NCP-1 for 2 h. Cells were washed and intracellularly stained with an anti-IgG Alexa Fluor 488 (dilution 1:500, Invitrogen; Cat# A11029) antibody for 30 min. Cells were washed and stained with Hoechst (dilution 1:1000, Invitrogen cat# H3570). Images were acquired with an Opera Phenix high content confocal microscope (PerkinElmer). The number of N- positive objects and nuclei were quantified using the Harmony Software v4.9 (PerkinElmer). The viral titer (Infectious units /mL) was calculated from the last positive dilution with 1 infectious unit (IU) being 3 times the background.

### Antibodies
Bamlanivimab, Casirivimab, Etesevimab, Imdevimab, Cilgavimab, Tixagevimab and Sotrovimab were provided by CHR Orleans. Bebtelovimab was produced as previously described[49]. NCP-1 antibody was selected from a series of mouse monoclonal antibodies (mAbs) directed against recombinant SARS-CoV-2 nucleoprotein. Four BALB/c mice were immunized by intraperitoneal injections at 3-week intervals of 50 µg of recombinant SARS-CoV-2 nucleoprotein mixed with alum adjuvant. The two mice presenting the best immune response were selected and were given a daily intravenous booster injection of SARS-CoV-2 nucleoprotein for three days. Two days after the last boost, hybridomas were produced by fusing spleen cells with NS1 myeloma cells. The Hybridoma culture supernatants were screened for the presence of anti-SARS-CoV-2 nucleoprotein antibodies with an ELISA checking their capacity to bind nucleoprotein biotin conjugate. Selected hybridomas were subsequently cloned and antibodies were produced from culture supernatants and purified by protein A affinity chromatography. All animal experiments were performed in accordance with the European Directive 210/63/ECC on the protection of animals used for scientific purposes and were approved by the Ethics Committee of the Commissariat à l'Energie Atomique (CEtEA "Comité d'Ethique en Expérimentation Animale" N°44) and by the French Ministry of Higher Education and Research under registration number APAFIS#3085-2015120909154560.

IGROV-1 cells were stained with goat anti-ACE2 polyclonal antibodies (AF933−R&D) 1:500 and mouse anti-TMPRSS2 antibodies (#HPA035787−Atlasantibodies) 1:500. Secondary antibodies coupled with Alexa Fluor 488 or 647 (Invitrogen) were used at 1:500. Staining was analyzed by flow cytometry on Attune Nxt Software v3.2.1.

### Statistical analysis
Flow cytometry data were analyzed using FlowJo v.10 (TriStar). Calculations were performed using Excel v16.46 365 (Microsoft). Figures were generated using Prism 9 (GraphPad Software). Statistical analysis was conducted using GraphPad Prism 9. Statistical significance between different groups was calculated using the tests indicated in each figure legend.

### Reporting summary
Further information on research design is available in the Nature Portfolio Reporting Summary linked to this article.

## Data availability
All data supporting the findings of this study are available within the article or from the corresponding authors upon reasonable request without any restrictions.

The raw data generated in this study are provided in the Source Data file.

The sequencing data generated in this study have been deposited in the GISAID database under accession code: D614G: EPI_ISL_414631; BA.1 ID: EPI_ISL_6794907; BA.5 ID: EPI_ISL_13660702; BA.2.75.2 ID: EPI_ISL_15731524; BQ.1.1 ID: EPI_ISL_15731523; BA.4.6 ID: EPI_ISL_15729633 Source data are provided with this paper.

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

## Acknowledgements

The authors thank the patients who participated to this study, the members of the Virus and Immunity Unit and other teams for discussion and help, Laurent Bélec, Jean-Louis Baillard and Julien Rodary (Hôpital Europeen Georges Pompidou) for organizing the collect and analysis of nasopharyngeal swabs. Nathalie Aulner and the staff at the UtechS Photonic BioImaging (UPBI) core facility (Institut Pasteur), a member of the France BioImaging network, for image acquisition and analysis, Elise Yang et Marc Plaisance (CEA) for anti-N antibody production, Fabienne Peira, Vanessa Legros, Aurelie Theillay, Sandra Pallay and Daniela Pires Roteia (CHR Orléans) for their help with the cohorts. The authors thank the KU Leuven University authorities and J. Arnout, B. Lambrecht, C. Van Geet, and L. Sels for their support. Work in OS lab is funded by Institut Pasteur, Urgence COVID-19 Fundraising Campaign of Institut Pasteur, Fondation pour la Recherche Médicale (FRM) EQU202003010172, ANRS, the Vaccine Research Institute (ANR-10-LABX-77), Labex IBEID (ANR-10-LABX-62-IBEID), ANR / FRM Flash Covid PROTEO-SARS-CoV-2, ANR Coronamito, HERA European funding, Sanofi and IDISCOVR. DP is supported by the Vaccine Research Institute. The E.S.-L. laboratory is funded by Institut Pasteur, the INCEPTION program (Investissements d'Avenir grant ANR-16-CONV-0005) and the French Government's Investissement d'Avenir program, Laboratoire d'Excellence 'Integrative Biology of Emerging Infectious Diseases' (grant no. ANR-10-LABX-62-IBEID). HERA European funding and the NIH PICREID (grant no U01AI151758). The Opera system was co-funded by Institut Pasteur and the Région ile de France (DIM1Health). Work in UPBI is funded by grant ANR-10-INSB-04-01 and Région Ile-de-France program DIM1-Health. The funders of this study had no role in study design, data collection, analysis, and interpretation, or writing of the article. P.M. acknowledges the support of a COVID-19 research grant from 'Fonds Wetenschappelijk Onderzoek'/ Research Foundation Flanders (grant G0H4420N) and 'Internal Funds KU Leuven' (grant 3M170314).

## Author contributions

Experimental strategy design, experiments: D.P., T.B., I.S., F.G.-B., F.P., L.G., and O.S. Vital materials: P.M., C.P., J.P., M.S., R.S., F.F., N.M., J.D., R.S., H.M., E.A., L.H., D.V., T.P., and H.P. Viral sequencing: P.M., M.P., S.M., J.P., E.S.-L., D.V., and H.P. Manuscript writing and editing: D.P., T.B., and O.S.

## Competing interests

T.B., C.P., H.M., and O.S. have a pending patent application for an anti-RBD mAb not used in this study (WO/2022/228827). All other authors have no conflict of interest.

## Additional information

[1]Virus and Immunity Unit, Institut Pasteur, Université Paris Cité, CNRS UMR3569 Paris, France. [2]Vaccine Research Institute, Créteil, France. [3]KU Leuven, Department of Microbiology, Immunology and Transplantation, Laboratory of Clinical and Epidemiological Virology, Leuven, Belgium. [4]G5 Evolutionary Genomics of RNA Viruses, Institut Pasteur, Université Paris Cité, Paris, France. [5]Humoral Immunology Laboratory, Institut Pasteur, Université Paris Cité, INSERM U1222 Paris, France. [6]Laboratoire de Virologie, Service de Microbiologie, Hôpital Européen Georges Pompidou, Paris, France. [7]Service d'accueil des urgences, Hôpital Européen Georges Pompidou, Paris, France. [8]Service de Pharmacologie et Immunoanalyse (SPI), CEA, INRA, Université Paris-Saclay, F-91191 Gif-sur Yvette, France. [9]Institute for Integrative Systems Biology (I2SysBio), Universitat de València-CSIC, 46980 Paterna València, Spain. [10]Department of Genetics, Universitat de València, València, Spain. [11]University Hospitals Leuven, Department of Laboratory Medicine, National Reference Centre for Respiratory Pathogens, Leuven, Belgium. [12]KU Leuven, Department of Microbiology, Immunology and Transplantation, Laboratory of Clinical Microbiology, Leuven, Belgium. [13]CHR d'Orléans, Service de Maladies Infectieuses, Orléans, France. [14]Functional Genomics of Solid Tumors (FunGeST), Centre de Recherche des Cordeliers, INSERM, Université de Paris, Sorbonne Université, Paris, France. [15]These authors contributed equally: Thierry Prazuck, Hélène Péré. [16]These authors jointly supervised this work: Delphine Planas, Olivier Schwartz. ✉e-mail: delphine.planas@pasteur.fr; olivier.schwartz@pasteur.fr

