## [Peer review file · Nature Communications]

REVIEWERS' COMMENTS

Reviewer #1 (Remarks to the Author):

Comments raised in the previous review (NMED-A123243, Reviewer 1) have been adequately addressed by the authors.

Minor: "d" in the legend of Figure 4 should be replaced by "c" as the figure does not contain a panel d.

Reviewer #2 (Remarks to the Author):

The manuscript by Planas has been amended and updated to respond to the reviewer comments, with a number of clarifications and corrections. Interestingly, the authors were able to provide some preliminary data that clarify the cellular pathways utilised by the IGROV-1 cells as requested by two of the three reviewers. The authors state they are reserving such data for future studies. While the manuscript is suitable for publication in its current form, the impact of the work would be significantly enhanced if the unique phenotype of these cells were mechanistically clarified.

Reviewer #3 (Remarks to the Author):

I am satisfied with the response.

REVIEWERS' COMMENTS

We thank the three reviewers their positive appreciation of our work.

Reviewer #1 (Remarks to the Author):

Comments raised in the previous review (NMED-A123243, Reviewer 1) have been adequately addressed by the authors.

Minor: "d" in the legend of Figure 4 should be replaced by "c" as the figure does not contain a panel d.

We have modified the legend of Figure and replaced "d" by "c".

Reviewer #2 (Remarks to the Author):

The manuscript by Planas has been amended and updated to respond to the reviewer comments, with a number of clarifications and corrections. Interestingly, the authors were able to provide some preliminary data that clarify the cellular pathways utilised by the IGROV-1 cells as requested by two of the three reviewers. The authors state they are reserving such data for future studies. While the manuscript is suitable for publication in its current form, the impact of the work would be significantly enhanced if the unique phenotype of these cells were mechanistically clarified.

We thank reviewer #2 for sharing our opinion that reporting the identification of the high sensitivity of IGROV-1 to Omicron will be useful to other groups. We agree that characterizing the underlying mechanisms may allow to rationally select sensitive cell lines and to understand the virological features of the Omicron family. This investigation is still ongoing. As mentioned in our previous response, our preliminary data suggest that the increased IGROV-1 sensitivity may be due to improved endocytic viral entry. To avoid delays in publication, we have decided not to include the results in the present article.

Reviewer #3 (Remarks to the Author):

I am satisfied with the response.